# A Community Waterborne *Salmonella* Bovismorbificans Outbreak in Greece

**DOI:** 10.3390/ijerph21020167

**Published:** 2024-02-01

**Authors:** Lida Politi, Kassiani Mellou, Anthi Chrysostomou, Georgia Mandilara, Ioanna Spiliopoulou, Antonia Theofilou, Michalis Polemis, Kyriaki Tryfinopoulou, Theologia Sideroglou

**Affiliations:** 1Department of Microbial Resistance and Infections in Health Care Settings, Directorate of Surveillance and Prevention of Infectious Diseases, National Public Health Organization, 15123 Athens, Greece; l.politi@eody.gov.gr; 2Directorate of Surveillance and Prevention of Infectious Diseases, National Public Health Organization, 15123 Athens, Greece; 3Department of Foodborne and Waterborne Diseases, Directorate of Surveillance and Prevention of Infectious Diseases, National Public Health Organization, 15123 Athens, Greece; a.chrysostomou@eody.gov.gr (A.C.); t.sideroglou@eody.gov.gr (T.S.); 4National Reference Centre for Salmonella and Shigella, School of Public Health, University of West Attica, 11521 Athens, Greece; gmandilara@uniwa.gr; 5Central Public Health Laboratory, National Public Health Organization, 16672 Vari, Greece; i.spiliopoulou@eody.gov.gr (I.S.); k.tryfinopoulou@gmail.com (K.T.); 6Water Microbiology Laboratory, Central Public Health Laboratory, National Public Health Organization, 16672 Vari, Greece; a.theofilou@eody.gov.gr; 7National Electronic Antimicrobial Resistance Surveillance Network, Central Public Health Laboratory, National Public Health Organization, 16672 Vari, Greece; m.polemis@eody.gov.gr

**Keywords:** waterborne, outbreak investigation, gastroenteritis, case–control study

## Abstract

Background: In August 2022, the Hellenic National Public Health Organisation was notified about a gastroenteritis outbreak in town A in Southern Greece. Investigations aimed to identify the source and implement control measures. Methods: Case definition categories were used in a 1:3 case–control study. Cases and controls were interviewed about various exposures. Cases’ stool samples were cultured on agar plates and characterised by serotyping, antimicrobial susceptibility testing and Pulse Field Gel Electrophoresis (PFGE). Environmental investigations included tap water sampling for microbiological and chemical analysis in town A and inspection of the water supply system. Results: We identified 33 cases (median age: 17 years). Tap water consumption was the only significant risk factor for gastroenteritis (OR = 5.46, 95% CI = 1.02–53.95). *Salmonella* (*S.*) Bovismorbificans isolated from eight stool and one tap water samples had identical PFGE profiles. No resistant isolates were identified. Residual chlorine levels were lower than the acceptable limits before and during the outbreak. We advised consumption of bottled water and adherence to strict hand hygiene rules until tap water was declared suitable for drinking. Conclusions: Epidemiological and molecular data revealed a waterborne *S.* Bovismorbificans outbreak in town A. We recommend local water safety authorities to ensure that residual chlorine levels comply with the legislation towards water safety planning, to mitigate risks.

## 1. Introduction

In Europe, progress has been noticed in wastewater treatment facilities and expansive pipeline networks that supply quality safe water to both cities and rural areas. However, poor system maintenance, infrastructure failures and natural disasters still occur. These often lead to outbreaks and reveal the serious effects of low-quality water (even short-term) on developed countries [1,2,3]. Monitoring mechanisms and surveillance for early detection are important for developed countries and should not be neglected [4,5,6,7]. Although uncommon in industrialized countries, community waterborne outbreaks (WBOs) may affect a large number of people in a short period of time, as contaminated drinking water reaches several households via the regional water supply system [8,9,10]. Detection of WBOs may be delayed and the number of affected people may be underestimated, due to the health-seeking behaviour of symptomatic people who may not seek professional healthcare and choose self-treatment by store-bought anti-diarrhoeal drugs [11]. As safe drinking water is a major public health concern, public drinking water systems are required to be disinfected, usually by chlorination, prior to distribution [12,13].

In Greece, despite public health measures, advances in implemented relevant protocols and legislation that ensure water quality and safety, WBOs continue to challenge the health care systems in different geographical regions [14,15,16,17]. According to the Hellenic National Public Health Organisation (EODY) surveillance data, 34 WBOs were recorded between 2004 and 2021. The largest outbreak was detected in 2019 in Northern Greece, with 638 recorded cases [15]. 

Here, we present the findings of a community waterborne outbreak investigation that occurred in Southern Greece in August 2022 aiming to depict challenges in management in a European country and stress the need for continuous efforts to assure the quality of drinking water.

### The Outbreak 

On 19 August 2022, EODY, was notified by the Health Care Centre (HCC) of town A (estimated population of 2000 people), of an increasing number of gastroenteritis cases. There were no reports on such cases from nearby areas. A gastroenteritis outbreak in town A, most likely started on 10 August 2022, was confirmed based on the findings of the preliminary investigation. During that period, apart from residents, many visitors stayed in town A for summer holidays.

The first reports of the microbiological analysis of stool samples from symptomatic cases confirmed the presence of *Salmonella* spp. Soon after the outbreak alert notification, an Outbreak Control Team (OCT) was established to further investigate the outbreak. Representatives from the regional Public Health Directorate (PHD), the Municipal Health Authorities (MHA), the National Reference Laboratory (NRL) for *Salmonella*, the Central Public Health Laboratory (CPHL) and the municipal Water and Sewage Authorities (WSA) were invited to participate in the OCT investigating the outbreak. 

The objectives of the investigation were the identification of the mode and vehicle of transmission and the determination of possible risk factors, hence the implementation of appropriate control measures. As of 24 August 2022, no new cases in town A were reported through case findings. 

## 2. Methods

Following the initial notification of increasing numbers of gastroenteritis cases in town A, an investigation commenced to reveal the source and mode of transmission, according to the steps of outbreak investigation. 

### 2.1. Active Case Finding

During the preliminary investigation it became apparent that cases had visited different Health Care Facilities (HCFs) in different towns and cities. Thus, active case finding was conducted with the cooperation of health care professionals of HCFs in town A, including the HCC of town A, in surrounding areas and in the Attica region, that also reported cases linked to the outbreak. The involved HCFs were asked to provide in a line list format (i) the total number of patients that were residents or visitors in town A with acute gastroenteritis symptoms as of 10 August onwards, (ii) their demographic characteristics (age, gender, etc.) and (iii) information on symptoms, day and time of symptom onset and hospitalisation status.

A sensitive case definition was initially formulated for active case finding and listing of all possible cases. A gastroenteritis case was defined as any resident or visitor of town A with ≥3 daily diarrhoeal episodes and one or more of the following symptoms: fever (≥38.0 °C), vomiting, nausea, or abdominal pain; they also had to have had an onset of symptoms from 10th of August 2022 onwards.

The contacted HCFs, private practices and laboratories were also asked to report any new salmonellosis cases (in case of isolation/detection of *Salmonella* spp. in stool samples) through the Mandatory Notification System. In Greece, the National Mandatory Notification System monitors 53 infectious diseases. According to this surveillance system, epidemiological data on every case of these diseases must be collected by Greek physicians and must be notified according to a pre-defined timetable, both at the Hellenic National Public Health Organization, which is the national competent body for the epidemiological surveillance, and at the Public Health Directorates of the corresponding prefectures, which are the competent local authorities, responsible for taking public health measures at the local level.

### 2.2. Study Design, Study Population, Case and Control Definitions

Preliminary interviews with cases did not identify any common exposure to activities, meals, or festivities during the fortnight prior to the probable outbreak onset. Thus, the main hypothesis formed was that this outbreak was potentially waterborne and that tap water in town A was the vehicle of transmission. An unmatched 1:3 case–control study was conducted to test this hypothesis and to identify risk factors for salmonellosis in town A in August 2022. For each identified case we selected 3 controls. 

The study population were all residents and visitors of town A, between 10–24 August 2022. Both cases and controls that had not been staying or visiting town A during the incubation period (6–72 h prior to symptom onset) or during the study period (10–24 August 2022), respectively, were excluded from the study. 

As further serotyping of the first stool samples indicated the presence of *Salmonella enterica* subsp. *enterica* ser. Bovismorbificans (6,8:r:1,5) (*S*. Bovismorbificans), this information was included in the formulation of a specific case definition. A confirmed outbreak case was defined as “any resident or visitor of town A with ≥3 daily diarrhoeal episodes and a stool sample positive for *S*. Bovismorbificans, from 10 to 24 of August 2022”. A probable case was defined as “every resident or visitor of town A with ≥3 daily diarrhoeal episodes and a stool sample positive for *Salmonella* spp., from 10 to 24 of August 2022”. A possible case was defined as “every resident or visitor of town A, with ≥3 daily diarrhoeal episodes, from 10 to 24 of August 2022”.

A control was defined as “any resident or visitor of town A between 10–24 August 2022 who did not exhibit symptoms of gastroenteritis but visited the HCC or the paediatricians (in case of children) in town A for other reasons, such as prescribing medicines, scheduled appointments for other pathological situations, diagnostic imaging examinations, COVID-19 testing, routine paediatric vaccination”.

### 2.3. Data Collection, Sample Size, and Statistical Analysis

The structured questionnaire included information on demographics (age, sex, residency, and so on.), date of symptom onset, type (vomiting, diarrhoea, fever, fatigue, abdominal pain, joint pain, and nausea) and duration of symptoms, need for and duration of hospitalisation, contact with other cases before or after symptom onset, participation in various social activities, consumption of certain food items and drinks, consumption of tap water or filter water or bottled water or other water source at home and at work, use of tap water for daily routine, such as bathing, washing and cooking. Moreover, as infants, i.e., children aged younger than 1 year and young children were reported among the cases, questionnaires included additional questions on dietary requirements and practices for infants and children, such as breastfeeding, milk types consumed, etc. Controls were interviewed about the same exposures. Interviews of cases and controls were conducted by phone calls, by both investigators of EODY and the regional PHD. As stool samples from cases were positive for *Salmonella* spp., by considering the maximum incubation period of the disease, all questions were related to the period of 3 days prior to symptom onset for cases and the period of 3 days prior to the visit to the HCC or paediatrician in town A for controls. For cases and controls younger than 14 years, questions were answered by their guardians.

EpiData Manager (v4.6.0.6.) and EpiData Entry Client (v4.6.0.6.) (The EpiData Association, Odense, Denmark) were used to create the database and enter data, respectively. Data analysis was conducted using Stata version 16 (StataCorp, College Station, TX, USA). Numeric variables were presented as means and standard deviations (SD) or as medians and interquartile ranges (IQR). Categorical variables were presented as absolute frequencies and percentages. Univariable analysis with the calculation of the chi-square test was used to test for associations between the categorical variables and the occurrence of illness. The outcome was defined as any confirmed, probable, or possible case. Odds ratios (OR) and the corresponding 95% confidence intervals (95% CI) were calculated. *p*-values less than 0.05 were considered statistically significant. Due to possible differences in exposures and difference in sources of recruitment of cases and controls, a further analysis by age category (>14 and ≤14 years old) was conducted. This age division was based on the fact that in Greece, people aged ≤14 are referred to paediatricians, while >14 are referred to pathologists.

### 2.4. Ethical Considerations

Collected data were anonymised, entered in a database by the epidemiologists of the OCT at the EODY’s premises, processed and analysed according to national and European Union laws. The study protocol was submitted to, and reviewed and approved (approval code and date: 3206/30 August 2022) by the Scientific Board of EODY. The study participants were informed on personal data process. Interviews with the study participants were conducted during the period 31 August 2022 to 20 September 2022. Furthermore, in terms of a publication in a scientific journal, a data request was submitted to, reviewed and approved (approval code and date: 6131/27 March 2023) by the Scientific Board of EODY. 

### 2.5. Μicrobiological Investigations—Clinical Samples

Stool samples from gastroenteritis outbreak cases in the involved HCFs were, directly and after 24 h enrichment, cultured on agar plates and screened for gastrointestinal pathogens. Serotyping of the available identified *Salmonella* isolates was performed at the NRL for *Salmonella* according to the White–Kaufmann–Le Minor Scheme [18,19]. Antimicrobial susceptibility testing of *Salmonella* isolates was performed using the disk diffusion method and results were evaluated according to the European Committee on Antimicrobial Susceptibility Testing breakpoints (EUCAST) [20]. Genotyping of seven of the outbreak *S*. Bovismorbificans isolates and one of the water sample isolates was performed using Pulsed Field Gel Electrophoresis (PFGE), after the digestion of genomic DNA with XbaI macrorestriction endonuclease, according to Standard Operating Procedures (SOPs) [21]. Two epidemiologically non-related clinical *S*. Bovismorbificans isolates (different temporal and geographical isolation) were also tested to ensure discriminatory power of PFGE on the specific serotype.

### 2.6. Environmental Investigation and Water Sampling

The regional and the local PHDs and the MHA were formally informed about the outbreak and asked to provide information on the town’s water supply system (maps of the water supply system within town A and surrounding areas), information of any maintenance activities on the water supply system over two month priors to the outbreak, available data on chemical and microbiological analysis of the water over the past two months (June and July 2022) prior to the outbreak and results of the inspection of the water supply system. Collected samples from case households, water intake tanks and various collection points of the water supply system were collected by the MHA and the local PHD and were tested for residual chlorine and the presence of various bacteria as required by the legislation [14]: total plate count at 22 °C and 37 °C, number of total coliforms colonies per 100 mL, number of *Enterococci* spp. colonies per 100 mL, number of *Escherichia coli* colonies per 100 mL, number of *Pseudomonas aeruginosa* colonies per 100 mL, number of *Salmonella* spp. colonies per 1000 mL, and number of *Clostridium perfringens* colonies per 100 mL. Water samples collected by the MHA prior and after the outbreak onset, as well as those collected by the local PHD on 11 May and 6 September 2022, were tested by two private collaborating laboratories (laboratory 1 and laboratory 2, respectively). Water samples collected by the local PHD on 24 and 25 August 2022 were tested at CPHL. One water sample collected by the parent of an outbreak case on 16 August 2022 from their household was tested at private laboratory 2. The structural integrity of the public water network was inspected by the local PHD. Information on the parameters tested for each sample by different entities are summarised in Table 1.

## 3. Results

### 3.1. Descriptive Epidemiology

From 10 to 24 August 2022, 40 cases of acute gastroenteritis, geographically distributed all over the town, were identified from the HCC of town A, two general hospitals in the nearby cities of town A, a private hospital in Athens, two specialised paediatric hospitals in Athens (where cases were referred to), two paediatricians (private practices) and two private microbiology laboratories in the area of town A) and one private microbiology laboratory in Athens. However, only 34 cases fulfilled the case definition and were included in the study. Likewise, from the original estimated number of controls (*n* = 102), only 81 accepted to be included in the study. Therefore, the final study population included 33 cases and 81 controls (response rate among cases and controls: 97.1% and 79.4%, respectively). 

Gender distribution among cases and controls was similar (55% and 56% of cases and controls, respectively, were females). There was no statistically significant difference in gender distribution among cases and controls (when the whole study population was considered: *p* = 0.548; for individuals > 14 years: *p* = 0.509; for individuals ≤ 14 years: *p* = 0.638; for children < 1 year: *p* = 0.654). 

The median age of cases and controls was 17 years (IQR: 60) and 24 years (IQR: 49), respectively. Not any statistically significant difference became apparent in age distribution among cases and controls in total (*p* = 0.968), in the age groups >14 years (*p* = 0.127) and in those younger than 1 year (*p* = 0.166). Among cases and controls aged ≤14 years (including children younger than 1 year), median age was 1 year (IQR: 5) and 6 years (IQR: 6), respectively (*p* = 0.002). 

Most common reported symptoms were diarrhoea (100%) and fever ≥ 38 °C (79%). The distribution of symptoms among cases and according to age distribution is depicted in Table 2. 

Twenty-six cases (79%) visited a HCF, among which thirteen (50%) were >14 years old. Seven cases (21%) were hospitalised, among which four (57%) were ≤14 years old. Three (75%) out of those four were children aged <1 year. The median duration of symptoms was 7 days. 

The distribution of cases by date of symptom onset shows a sharp increase in cases from 10th of August 2022, reaching a peak on 13rd of August 2022 (Figure 1). Following that, a drop in the number of cases was observed, with a few cases reported between the 16th and 24th of August 2022. 

### 3.2. Analytical Epidemiology

According to the univariable analysis for cases and controls >14 years old, there was a statistically significant association between consumption of tap water and illness: the odds of tap water consumption among cases were 5.5 times higher than the odds of tap water consumption among controls (OR = 5.46, 95% CI = 1.02–53.95, *p* = 0.025). The consumption of bottled water seemed to have a protective effect (OR = 0.18, 95% CI = 0.03–0.8, *p* = 0.009). Although the study participants were asked about the amount of daily tap water consumption, an association between the aforementioned variable and the risk of illness did not become apparent. Finally, no food consumption or common activity had a statistically significant association with the occurrence of gastroenteritis symptoms. The results of the univariable analysis for cases and controls aged 14 years or below including infants did not reveal any statistically significant associations between possible exposures and the occurrence of gastroenteritis. The analysis was repeated for infants aged ≤1-year cases and controls only. Again, no statistically significant associations were revealed. Results are depicted in Table 3 and Table 4. 

### 3.3. Environmental Investigation and Water Sampling Results

Town A is served by one public water supply system, receiving water from two functional water tanks (tank A and tank B), which receive water from three underground water intakes. The tanks are covered and secured. One main water pipe exits the one tank and connects with three pipes of the second water tank. The water pipe system splits further down to a system of five smaller water pipes that supply households of town A. The water is disinfected via chlorination. Mapping of the water supply system was not available to the OCT.

Inspection of the water supply system on 31 August 2022 revealed that there was a plot used for grazing sheep in the catchment area close to the water intakes (15 m). However, there were no animals in that plot during inspection, or signs of recent use of the plot. Only minor leakages were identified in the functional water tanks, but these were maintained daily. The integrity of the piping system was intact and there was no sign of contamination of the water supply from sewage overflow. Residual chlorine records were not presented during inspection, nor for the period prior to, neither during the event.

A total of 21 tap water samples from case households, water intake tanks and various collection points of the water supply system were collected by the local PHD and municipal WSA. Specifically, three samples were collected prior to the outbreak and 17 samples were collected after the outbreak onset, of which four samples during the outbreak period and 13 samples after the last notified outbreak case. Finally, one tap water sample collected during the outbreak period by the parent of an outbreak case tested positive for *Salmonella* spp. in a private laboratory and serotyped at NRL as *S.* Bovismorbificans. It should be noted that not all samples were tested for all indicators. 

Residual chlorine levels were tested in a total of 15 samples, collected by the local PHD and municipal WSA. A total of 3 samples were collected prior to the outbreak and 12 collected after the outbreak onset. In all 15 samples, residual chlorine measurements were lower than the minimum value (<0.2 mg/L) required by the relevant Greek legislation.

One tap water sample collected by the local PHD a day after the last notified outbreak case and tested at CPHL was positive for *Salmonella* spp., which was serotyped at the NRL for *Salmonella* as *Salmonella enterica* subsp. *enterica* ser. Bardo (8:e,h:1,2).

Table 1 summarizes the results of water samples testing per each parameter and authority prior, during and after the outbreak, while Figure 2 illustrates all the steps of the investigation.

### 3.4. Laboratory Investigation Results—Clinical Samples

Laboratory investigations confirmed the presence of *Salmonella* spp. in stool samples of 15 (45%) cases. Eight *Salmonella* spp. isolates were sent to the NRL for serotyping. All of them serotyped as *S*. Bovismorbificans (6,8:r:1,5). All *S.* Bovismorbificans outbreak-associated human and water isolates presented a coincident antibiotype (all susceptible to the tested antimicrobial agents) and indistinguishable PFGE patterns. The two epidemiologically non-related *S.* Bovismorbificans isolates presented a very different molecular profile (Figure 3). 

## 4. Discussion

Our epidemiological investigation indicated that an outbreak of non-typhoidal salmonellosis occurred in a small town A in Southern Greece during 10–24 August 2022. The results of the case–control study, along with the fact that no common activities of the cases were reported prior to the outbreak, are supportive of the waterborne origin of the outbreak. The fact that genotyping revealed indistinguishable PFGE patterns in human isolates suggest that this was a common-source *S*. Bovismorbificans outbreak. It has to be noted that *S*. Bovismorbificans is the fourth most common serotype in clinical isolates in Greece although in very low percentages (3.4% for the period 2003–2020) [22]. In 2022, *S*. Bovismorbificans was identified in 11.4% of the clinical serotyped *Salmonella* spp. isolates, according to NRL for Salmonella unpublished data. Furthermore, clinical isolates presented an indistinguishable PFGE profile with that of the isolate of the water that was sampled by the parent of an outbreak case. 

The duration of the outbreak was rather short (15 days) and the number of cases was rather low for a community outbreak [8,10]. This low number of identified cases after active case finding is irregular, as waterborne outbreaks usually affect large numbers of people of all age groups within a short time period. Perhaps the choice of residents to consume bottled water instead of tap water, along with the authorities’ advice to practice strict hand hygiene and the chlorination of the water supply system that probably took place during the event, have led to the rapid containment of the outbreak. 

It is not, however, the first time that the number of recorded cases is low in a WBO. In a waterborne *Salmonella* Typhimurium outbreak in the Chania area, on Crete Island, during February–March 2004, the number of recorded cases was low (*n* = 37), as well as in a *Campylobacter jejuni* waterborne outbreak in the same area on Crete Island, in 2009 (*n* = 37) [15,23]. A probable explanation is that a possible damage in the water supply system could have affected a small number of people at a certain point of time. A reason that could explain the small size of the abovementioned outbreaks, including the *S.* Bovismorbificans outbreak that we described in this paper, is the limited persistence of bacteria in the water supply system in comparison to viruses. During WBOs caused by viruses, the detection period for the infective stage in the water at 20 °C may extend the period of one month [24]. 

Enteric pathogens, such as *Salmonella* spp., are not commonly isolated in drinking water systems. They can, however, be introduced into these systems through stormwater runoff, sewage overflow or animal waste contamination, causing waterborne non-typhoidal salmonellosis [25,26]. In this outbreak, the investigation did not reveal the way the water might have been contaminated with *Salmonella* spp. Although the coordination of the investigation took place immediately after the alert, the notification was not timely. The fact that cases visited many different HCFs made active case finding and epidemiological investigations time-consuming. These delays may have led to the failure of prompt environmental investigations in revealing how the water supply system was contaminated. Furthermore, the delayed inspection of the water network combined with the lack of mapping of the water supply network did not allow for the investigators to conclude on the origin of a possible faecal contamination of the water supply. The low number of tap water samples collected by both competent authorities during the outbreak period resulted in less-than-optimum representativeness of the sampling points of the water network. In addition, not all samples were tested for *Salmonella*, faecal indicator bacteria and residual chlorine, according to Greek legislation. This demonstrates the need that sampling authorities should be better informed about the required volume of the laboratory sample for each analysis and/or have an earlier consultation with the laboratory, to collect samples of sufficient volume for the performing of all desired parameters. Although not all water samples were tested for residual chlorine, the fact that it was lower than the minimum value required by the relevant Greek legislation may suggest deficiencies in water sanitation practices. Finally, the absence of chlorination records, prior and during the event, as well as during inspection, did not shed light to any probable chlorination deficiencies that could have led to the persistence of *Salmonella* spp. in the water supply system. 

Regarding the tap water sample collected by the parent of an outbreak case, SOPs may have not been followed during sampling, since it was not performed by an official authority. However, the identification of *S.* Bovismorbificans in both tap water sample and cases’ stool samples, the presence of *S.* Bardo in tap water samples collected by authorities and the identification of a grazing sheep plot in the catchment area close to the water intakes enhance the hypothesis of a possible animal faecal contamination of the water supply system [27].

As cases occurred both among children and adults, a matched case–control design was first considered. However, for children younger than 14 years, neither the individual nor frequency matching of controls was feasible, due to the very low number of available representative controls. Thus, the unmatched case–control study did not assist the identification of risk factors in this age group, to strengthen the finding of the study in participants over 14 years.

As the majority of WBOs in Greece have been reported in decentralized areas, this outbreak investigation demonstrates the importance of timely notification of any increase in gastroenteritis cases in the community and of development of SOPs for water sampling [11,15,16,17]. Implementation of SOPs is a need that has been demonstrated several times during similar investigations in the past [15,17]. Apart from that, the ageing of water infrastructures in the outbreak areas should be taken into consideration and the need for water safety plan for the protection of water supplies is more evident than ever [28]. According to the revised European Directive 2020/2184, the quality of water intended for human consumption should protect public health from the adverse effects of contaminants. Among the main innovations laid out is the risk-based approach, in accordance with the principles of the Water Safety Plan generated by the World Health Organization (WHO) [29]. This need was heeded by the Greek legislation and relevant changes have been decided recently towards this direction [12,14]. The adoption of a risk-based preventive approach in drinking water surveillance, from the source to the tap, will enable hazard identifications, risk assessment and risk management for every different water supply system, adopting a zero-pollution action plan.

## 5. Conclusions

The investigation of this outbreak managed to provide both epidemiological and laboratory evidence on its’ possible source. Both analytical study and laboratory investigation of clinical and water samples indicated that the vehicle of this community *S*. Bovismorbificans outbreak was tap water. Thus, although not typical, this event could be considered as an outbreak of a common source over a short exposure period. 

Even though drinking water management policies in the past few decades have led to decreased disease burden, further strengthening of related policies is needed to address the remaining burden attributed to catchment and distribution realm-associated deficiencies and to groundwater viral and disinfection-only system outbreaks.

## Figures and Tables

**Figure 1 ijerph-21-00167-f001:**
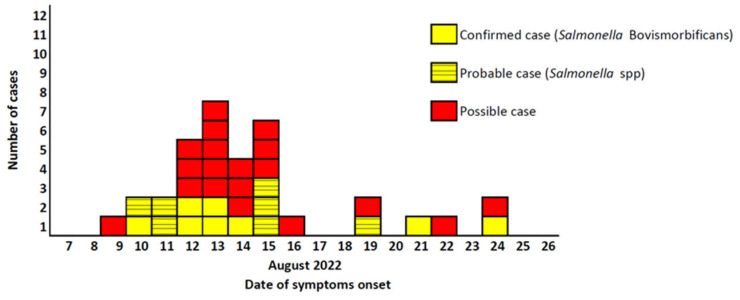
Distribution of confirmed, possible and probable non-typhoidal salmonellosis cases, by date of symptom onset among residents and visitors of town A (*n* = 33), Southern Greece, 10–24 August 2022.

**Figure 2 ijerph-21-00167-f002:**
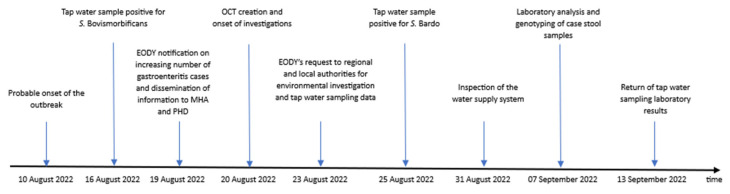
Timeline of events of the investigation of non-typhoidal salmonellosis cases of town A, Southern Greece, 10–24 August 2022.

**Figure 3 ijerph-21-00167-f003:**
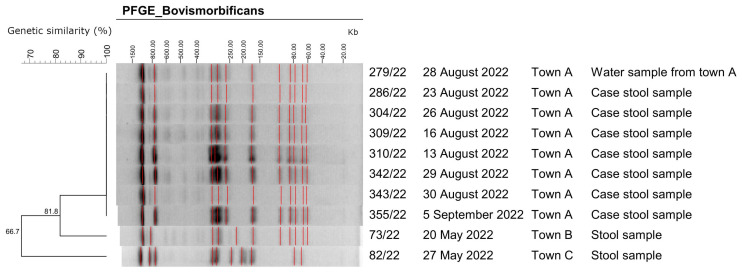
Representative genotypes of Xba-I digested genomic DNA of *Salmonella* Bovismorbificans isolates. Isolate 279/22 corresponds to a tap water sample (non-human origin), while isolates 73/22 and 82/22 correspond to samples that have no epidemiological link to the outbreak samples of town A. The dendrogram was constructed by using the program Bionumerics 6.6 (Applied Maths). Towns B and C are towns in different regions from Town A and are served by a different water supply system. The red lines mark the assigned bands of the gels.

**Table 1 ijerph-21-00167-t001:** Parameters tested for each tap water sample collected per authority, town A, Southern Greece, May–September 2022.

Indicators/ Pathogens	Municipal Water and Sewage Authority			Local Public Health Directorate							Parent of an Outbreak Case
	12 July 2022 1 Sample	3 August 2022 1 Sample	22 August 2022 1 Sample	11 May 2022 1 Sample	24 August 2022 3 Samples			25 August 2022 2 Samples		6 September 2022 11 Samples	16 August 2022 1 Sample
*Salmonella* spp./L	-	-	√	-	√	-	-	*S*. Bardo	-	√	*Salmonella* Bovismorbificans
*Escherichia coli* (cfu/100 mL)	√	√	√	√	-	√	√	-	√	√	-
*Clostridium perfringens* (cfu/100 mL)	√	√	-	-	-	-	-	-	-	√	-
*Pseudomonas aeruginosa* (cfu/100 mL)	√	√	-	-	-	-	-	-	-	-	-
Total coliforms (cfu/100 mL)	√	√	-	5	-	√	√	-	√	9 samples:0 2 samples: 15 and 35	-
Enterococci (cfu/100 mL)	√	√	√	√	-	√	√	-	√	√	-
Total plate count 22 °C (cfu/mL) Accepted limit: <1 cfu/100 ml	15	5	8	12	-	10	4	-	√	<3 to >300	-
Total plate count 37 °C (cfu/mL) Accepted limit: <1 cfu/100 ml	9	<1	14	20	-	<4	0	-	√	4 to >300	-
Residual chlorine (mg/lt). Accepted limit: >0.5 mg/lt	0.06	0.12	0.02	0.17	-	-	-	-	-	0.06–0.13	-

√ Compliance according to legislation. - Not tested for the respective parameter.

**Table 2 ijerph-21-00167-t002:** Distribution of symptoms among gastroenteritis cases by age category, town A, Southern Greece, 10–24 August 2022.

Symptoms	Number (%) of Cases > 14 Years Old	Number (%) of Cases ≤ 14 Years Old (Inc. < 1 Year Old)
Diarrhoea	15 (100.0%)	18 (100.0%)
Fever > 38 °C	11 (73.3%)	15 (83.3%)
Fatigue	11 (73.3%)	9 (50.0%)
Abdominal pain	8 (53.3%)	10 (55.6%)
Anorexia	5 (33.3%)	11 (61.1%)
Vomiting	6 (40.0%)	6 (33.3%)
Nausea	4 (26.7%)	4 (22.2%)
Joint pain	5 (33.3%)	1 (5.6%)

**Table 3 ijerph-21-00167-t003:** Univariable analysis on associations of food items and illness among cases and controls > 14 years old, town A, Southern Greece, 10–24 August 2022.

	Cases			Controls					
Food and Beverage Exposures	Total	Exposed	%	Total	Exposed	%	Odds Ratio	Confidence Interval	*p* Value
Tap water	15	13	86.67	46	25	54.35	5.46	[1.02–53.95]	0.025
Icecream	15	3	20.00	46	35	76.09	0.07	[0.01–0.31]	0.000
Milk	14	0	0.00	46	19	41.30	0.00	[0.00–0.32]	0.001
Pasta/rice	14	6	42.86	46	42	91.30	0.06	[0.00–0.68]	0.002
Soda	15	1	6.67	46	23	50.00	0.07	[0.00–0.56]	0.003
Bottled water	15	3	20.00	46	26	56.52	0.19	[0.03–0.87]	0.014
Egg	14	4	28.57	46	27	58.70	0.21	[0.04–0.91]	0.015
Seafood	15	0	0.00	46	9	19.57	0.00	[0.00–0.87]	0.036
Tap water for juice dilution	15	6	40.00	46	7	15.22	3.71	[0.80–16.46]	0.042
Red meat (beef/pork)	15	6	40.00	45	33	73.33	0.29	[0.07–1.22]	0.043
Beer	15	4	26.67	46	26	56.52	0.28	[0.06–1.15]	0.045
Chicken	15	3	40.00	46	32	69.57	0.40	[0.10–1.64]	0.135
Tap water for washing fruit	14	14	100.00	46	40	86.96	.	[0.50–.]	0.154
Fresh vegetables	14	12	85.71	45	43	95.56	0.28	[0.02–4.34]	0.201
Canned food	15	0	0.00	46	2	4.35	0.00	[0.00–2.92]	0.237
Dessert	14	2	14.29	46	14	30.43	0.47	[0.07–2.14]	0.281
Tap water for ice cube preparation	15	9	60.00	46	34	73.91	0.53	[0.13–2.23]	0.305
Cured meat	15	2	13.33	46	10	21.74	0.44	[0.04–2.43]	0.308
Use of water filter for tap water	14	3	21.43	44	5	11.36	2.13	[0.28–12.89]	0.341
Vodka	15	0	0.00	46	2	4.35	0.00	[0.00–6.09]	0.412
Spring water	15	0	0.00	46	2	4.35	0.00	[0.00–6.09]	0.412
Use of dishwasher	15	5	33.33	46	10	21.74	1.59	[0.35–6.54]	0.471
Whisky	15	0	0.00	46	1	2.17	0.00	[0.00–.]	0.565
Tap water use for brushing teeth	15	15	100.00	46	45	97.83	.	[0.00–.]	0.565
Tap water use for vegetable washing	15	15	100.00	46	45	97.83	.	[0.00–.]	0.565
Tsipouro (Greek alcoholic drink)	15	2	13.33	46	4	8.70	1.62	[0.13–12.73]	0.600
Tap water for baby cream preparation	8	1	12.50	40	3	7.50	1.76	[0.03–25.67]	0.640
Wine	15	3	20.00	46	7	15.22	1.39	[0.20–7.35]	0.664
Snack/drink	15	2	1333	46	6	13.04	0.73	[0.07–4.39]	0.712
Fish	15	4	26.67	46	15	32.61	1.14	[0.28–4.34]	0.833
Bathing with tap water	15	15	100.00	46	46	100.00	.	[0.00–.]	.

. Not defined.

**Table 4 ijerph-21-00167-t004:** Univariable analysis on associations of food items and illness among cases and controls ≤ 14 years old, town A, Southern Greece, 10–24 August 2022.

	Cases			Controls					
Food and Beverage Exposures	Total	Exposed	%	Total	Exposed	%	Odds Ratio	Confidence Interval	*p* Value
Tap water	18	8	44.44	34	16	47.06	0.90	[0.24–3.28]	0.857
Icecream	18	2	11.11	34	28	82.35	0.03	[0.00–0.17]	0.000
Dessert	18	1	5.56	34	20	58.82	0.04	[0.00–0.34]	0.000
Chicken	18	6	33.33	34	28	82.35	0.11	[0.02–0.47]	0.000
Tap water for juice dilution	18	2	11.11	34	21	61.76	0.08	[0.01–0.43]	0.000
Fish	18	1	5.56	34	18	52.94	0.05	[0.00–0.43]	0.001
Fresh vegetables	18	10	55.56	34	32	94.12	0.08	[0.01–0.51]	0.001
Egg	18	5	27.78	34	25	73.53	0.14	[0.03–0.58]	0.001
Cured meat	18	3	16.67	34	21	61.76	0.12	[0.02–0.58]	0.002
Seafood	18	0	0.00	34	12	35.29	0.00	[0.00–0.42]	0.004
Soda	18	0	0.00	34	12	35.29	0.00	[0.00–0.42]	0.004
Pasta/rice	18	13	72.22	34	32	94.12	0.16	[0.01–1.19]	0.028
Tap water for brushing teeth	18	13	72.22	34	32	94.12	0.16	[0.01–1.19]	0.028
Tap water for vegetable washing	18	16	88.89	34	34	100.00	0.00	[0.00–0.98]	0.047
Tap water for fruit washing	18	13	72.22	34	31	91.18	0.25	[0.03–1.56]	0.072
Red meat (beef/pork)	18	13	72.22	34	30	88.24	0.35	[0.06–1.94]	0.146
Tap water for baby cream preparation	18	3	16.67	34	2	5.88	3.20	[0.32–41.09]	0.209
Tap water for ice cube preparation	18	8	44.44	34	20	58.82	0.56	[0.15–2.06]	0.322
Snack/drink	18	1	5.56	34	5	14.71	0.34	[0.01–3.49]	0.326
Use of dishwasher	18	4	22.22	34	12	35.29	0.52	[0.10–2.23]	0.331
Milk	18	15	83.33	34	31	91.18	0.48	[0.06–4.11]	0.400
Canned food	18	0	0.00	34	1	2.94	0.00	[0.00–.]	0.463
Bottled water	18	14	77.78	34	14	70.59	1.46	[0.33–7.54]	0.578
Use of water filter for tap water	18	3	16.67	34	6	17.65	0.93	[0.13–5.17]	0.929
Bathing with tap water	18	17	94.44	34	34	100.00	.	[0.00–.]	.

. Not defined.

## Data Availability

Data is unavailable due to privacy and ethical restrictions.

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
