# Peer review of "A Community Waterborne Salmonella Bovismorbificans Outbreak in Greece"

_ijerph, 2024, doi:10.3390/ijerph21020167_

Round 1

Reviewer 1 Report

Comments and Suggestions for Authors

To authors;

The study analyses an outbreak of Salmonella bovimorbificans in a restricted area. The data have been meticulously analysed, well documented and presented. My suggestions for minor corrections are as follows. 

* Line 72: The "outbreak" section is mentioned in the introduction. However, it should also be included normally in the material and method.

* Line 108: Please add a short descriptive note about the "Mandatory Notification System" (national, real-time, etc.).

* Line 151: Please add EpiDataxxxx developer company, state, or internet connection.

* Line 170: If enrichment medium (GN broth etc?) was used for stool samples before culture, please specify. 

* The controls in Tables 3 and 4 are not understood. However, Line 382 mentions this issue. Please add the detail about the configuration/selection of control cases.

* If there are no legal restrictions, I suggest that a schematised case/control map (with chlorine levels etc) of the sampling area be included in the supplement.

Author Response

Dear reviewer, thank you for your remarks. Please find below our answers to your suggestions.

Reviewer 1

* Line 72: The "outbreak" section is mentioned in the introduction. However, it should also be included normally in the material and method.

Dear reviewer, thank you for your comment. Please find the added text through lines 90-92.

The added text is the following: Following the initial notification of increasing numbers of gastrenteritis cases in town A, an investigation commenced to reveal the source and mode of transmission, according to the steps of outbreak investigation.

* Line 108: Please add a short descriptive note about the "Mandatory Notification System" (national, real-time, etc.).

Dear reviewer, thank you for your comment. Please find the added text through lines 111-118.

The added text is the following: . In Greece, the National Mandatory Notification System monitors 53 infectious diseases. According to this surveillance system, epidemiological data on every case of these diseases must be collected by Greek physicians and must be notified according to a pre-defined timetable, both at the Hellenic National Public Health Organization, which is the national competent body for the epidemiological surveillance, and at the Public Health Directorates of the corresponding prefectures, which are the competent local authorities, responsible for taking public health measures at the local level.

* Line 151: Please add EpiDataxxxx developer company, state, or internet connection.

Dear reviewer, thank you for your comment. Please find the added text through lines 162-163 (The EpiData Association, Odense, Denmark).

* Line 187: If enrichment medium (GN broth etc?) was used for stool samples before culture, please specify. 

Dear reviewer, thank you for your comment. Stool samples from gastroenteritis outbreak cases in the involved HCFs were, directly and after 24-hour enrichment, cultured on agar plates and screened for gastrointestinal pathogens (lines 185-187).

* The controls in Tables 3 and 4 are not understood. However, Line 382 mentions this issue. Please add the detail about the configuration/selection of control cases.

Dear reviewer, thank you for your comment. We conducted a 1:3 case-control study. For each identified case we selected 3 controls. Due to possible differences in exposures and difference in sources of recruitment of cases and controls, a further analysis by age category (>14 and ≤ 14 years old) was conducted. This age division was based on the fact that in Greece people aged ≤14 are referred to paediatricians, while >14 to pathologists. The ratio 1:3 was met in age group >14 years old, but not in that of ≤14 years old, due to the fact that it was more difficult to recruit controls in the latter age group. However, the power of analysis was not substantially affected.

Please find the added text

- in “Study design, study population, case and control definitions” section in line 125:

For each identified case we selected 3 controls.

- in “Data collection, sample size, and statistical analysis” section, in lines 171-174:

Due to possible differences in exposures and difference in sources of recruitment of cases and controls, a further analysis by age category (>14 and ≤ 14 years old) was conducted. This age division was based on the fact that in Greece, people aged ≤14 are referred to paediatricians, while >14 to pathologists.  

* If there are no legal restrictions, I suggest that a schematised case/control map (with chlorine levels etc) of the sampling area be included in the supplement.

Dear reviewer, thank you for your comment. As stated in the limitation section, such information was requested but not provided. Therefore, such a map, also due to legal restrictions, could not be generated and included.

Reviewer 2 Report

Comments and Suggestions for Authors

The author's present an interesting study regarding drinking water quality and outbreak data, but I do have some recommendations for revisions. The title should be shortened and the introduction needs to be elaborated. Why do these outbreaks still occur? Provide information regarding other outbreaks? What are the impacts of these outbreaks?

The title is too long. Please shorten it. 

Line 38: Please revise the sentence since "off acceptable limits" does not make sense. 

Line 41: Please italicize the pathogen name. 

Line 42: The last sentence needs grammatical editing. What do you mean by systematically? A more specific concluding sentence would improve the abstract. 

Line 48: A single sentence should not be a paragraph. Please elaborate on this statement. Use a different word than "poor" for poor water quality. What are the serious effects on developed countries?

Line 65: unpublished data should be included in the reference, not listed in the sentence

Lines 110-115: Some of the word choices are not ideal (e.g., vehicle of the outbreak). Please revise this paragraph to utilize more appropriate vocabulary. Do you have any other information regarding the sample size needed to meet this ratio?

Please refrain from using "etc." too frequently in your manuscript.

Line 196: Is it Salmonella spp. colonies per 100mL or 1000mL?

Lines 227-229: Please update this sentence for grammatical clarity. 

Why are some of the p-values in bold in Table 3?

The discussion is well written. 

Comments on the Quality of English Language

I would recommend minor to moderate editing for the English language. 

Author Response

Dear reviewer, thank for your comments. Please fin below the answers to your suggestions.

Reviewer 2

*The title is too long. Please shorten it. 

Dear reviewer, thank you for your comment. Please find the altered text through lines 2-4. The new shorter version of the title is the following:

“The right to safe drinking water concerns developed countries too. A Salmonella Bovismorbificans waterborne outbreak in Greece, August 2022”.

*Line 38: Please revise the sentence since "off acceptable limits" does not make sense. 

Dear reviewer, thank you for your comment. Please find the altered text through lines 36-37. In general, we intended to mean that chlorination levels were below acceptable limits before and during the outbreak.

The altered text is the following: Residual chlorine levels were lower than the acceptable limits before and during the outbreak.

*Line 41: Please italicize the pathogen name. 

Dear reviewer, thank you for your comment. The pathogen name is italicized. The “S” is in italics, The serotype however (Bovismorbificans), should not be italicized, according to the Salmonella nomenclature.

*Line 42: The last sentence needs grammatical editing. What do you mean by systematically? A more specific concluding sentence would improve the abstract. 

Dear reviewer, thank you for your comment. We have altered the text according to your suggestions (lines 39-41).

The altered text is the following: We recommend local water safety authorities to ensure that residual chlorine levels comply with the legislation, towards water safety planning, to mitigate risks.

*Line 48: A single sentence should not be a paragraph. Please elaborate on this statement. Use a different word than "poor" for poor water quality. What are the serious effects on developed countries?

Dear reviewer, thank you for your comment. Please find the altered text through lines 46-50.

The altered text is the following: In Europe, progress has been noticed in wastewater treatment facilities and expansive pipelines network that supply quality safe water to both cities and rural areas. However, poor system maintenance, infrastructure failures, and natural disasters still occur. These often lead to outbreaks and reveal the serious effects of low-quality water (even short-term) on developed countries [1-3]. 

*Line 65: unpublished data should be included in the reference, not listed in the sentence.

Dear reviewer, thank you for your comment. We meant to say that EODY maintains all surveillance data, however, doesn’t publish everything. In order to avoid confusion, we have corrected the sentence (line 63).

The corrected text is as follows: According to Hellenic National Public Health Organisation (EODY) surveillance data, 34 WBOs were recorded between 2004 and 2021;

*Lines 110-115: Some of the word choices are not ideal (e.g., vehicle of the outbreak). Please revise this paragraph to utilize more appropriate vocabulary. Do you have any other information regarding the sample size needed to meet this ratio?

Dear reviewer, thank you for your comment. Please find the altered text through (line123: vehicle of transmission, line 124: for salmonellosis).

As far as the sample size is concerned: we initially wanted to conduct a matched case-control study that would be most effective for such a diverse case population (cases that may have different dietary needs according to their age). This, however, was not feasible, because there were no available controls for age matching. For that reason, we decided to go for an unmatched case-control study. We chose the ratio 1:3 because it is the most appropriate ratio; statistical confidence can be increased by taking more than one control per case. There is, however, a law of diminishing returns, and it is usually not worth going beyond a ratio of four or five controls to one case.

*Please refrain from using "etc." too frequently in your manuscript.

Dear reviewer, thank you for your comment. We have altered the text according to your suggestion.

*Line 210: Is it Salmonella spp. colonies per 100mL or 1000mL?

Dear reviewer, thank you for your comment. For Salmonella spp. colonies it is per 1000mL (line 211).

*Lines 227-229: Please update this sentence for grammatical clarity. 

Dear reviewer, thank you for your comment. Please find the altered text through lines 236-240.

The altered text is: Gender distribution among cases and controls was similar (55% and 56% of cases and controls, respectively, were females). There was no statistically significant difference in gender distribution among cases and controls (when the whole study population was considered: p=0.548; for individuals >14 years: p=0.509; for individuals ≤14 years: p=0.638; for children <1 year: p=0.654). 

*Why are some of the p-values in bold in Table 3?

Dear reviewer, thank you for your comment. We decided to make bold the ORs, CIs and p values that highlighted the statistically significant results of interest (tap water and bottled water consumption), to point them out for the readers.

Reviewer 3 Report

Comments and Suggestions for Authors

1. Please clarify the symptoms of Salmonella infection in this study. Are there different levels of symptoms assessed in the questionnaire?

2. Please describe how you collected accurate data. Did you filter out biased or unimportant data from the respondents who were not adequately informed

3. Could you please differentiate between Town A, Town B, and Town C in terms of water consumption and factors affecting the outbreak of Salmonella?

Author Response

Dear reviewer, thank you for your comments. Please find below the answers to your suggestions.

Reviewer 3

  1. Please clarify the symptoms of Salmonella infection in this study. Are there different levels of symptoms assessed in the questionnaire?

Dear reviewer, thank you for your comment. The symptoms can be found through lines 146-147: (vomiting, diarrhoea, fever, fatigue, abdominal pain, joint pain, nausea).

In the questionnaire, we had levels of symptoms for the following: for diarrhoea, we would ask cases how many episodes per 24 hrs they had. For vomiting, we would ask cases how many times within 24 hrs. For fever, we would ask cases what their temperature was (over 38°C). However, this information was not further analysed, due to the fact that cases could not point the exact number of diarrhoeas and vomiting episodes.

  1. Please describe how you collected accurate data. Did you filter out biased or unimportant data from the respondents who were not adequately informed

Dear reviewer, thank you for your comment. Cases and controls were already interviewed by the treating doctors in the healthcare facilities they visited. Therefore, we could cross-check the answers they gave us concerning some questions. Moreover, we had some questions in the questionnaire through which we would ask the same thing in a different manner. When inconsistencies were observed, we would call them back to clarify everything. Of course, recall bias can not be totally avoided in any study.

  1. Could you please differentiate between Town A, Town B, and Town C in terms of water consumption and factors affecting the outbreak of Salmonella?

Dear reviewer, thank you for your comment. Towns B and C are other towns in other regions of Greece where Salmonella Bovismorbificans isolates have been isolated. These other towns are served by a different water supply system.

You may find the added text through lines 334-336: Towns B and C are towns in different regions from Town A and are served by a different water supply system.

Round 2

Reviewer 2 Report

Comments and Suggestions for Authors

The title of the paper still needs to be revised. It is too long and descriptive. There are still typos (unnecessary periods and commas) that need to be corrected). Overall, the authors addressed my previous comments. 

Comments on the Quality of English Language

minor editing requested

Author Response

Dear reviewer, thank you for your comments. Please find below the altered title:

 "A community waterborne Salmonella Bovismorbificans outbreak in Greece".

Moreover, we have proceeded and removed any unnecessary commas and periods throughout the whole manuscript.